# Functionnectome as a framework to analyse the contribution of brain circuits to fMRI

Victor Nozais [1,2 ✉], Stephanie J. Forkel[1,2,3], Chris Foulon [4], Laurent Petit[1] &
Michel Thiebaut de Schotten [1,2 ✉]

In recent years, the field of functional neuroimaging has moved away from a pure localisationist approach of isolated functional brain regions to a more integrated view of these regions within functional networks. However, the methods used to investigate functional networks rely on local signals in grey matter and are limited in identifying anatomical circuitries supporting the interaction between brain regions. Mapping the brain circuits mediating the functional signal between brain regions would propel our understanding of the brain's functional signatures and dysfunctions. We developed a method to unravel the relationship between brain circuits and functions: The Functionnectome. The Functionnectome combines the functional signal from fMRI with white matter circuits' anatomy to unlock and chart the first maps of functional white matter. To showcase this method's versatility, we provide the first functional white matter maps revealing the joint contribution of connected areas to motor, working memory, and language functions. The Functionnectome comes with an open-source companion software and opens new avenues into studying functional networks by applying the method to already existing datasets and beyond task fMRI.

[1] Groupe d'Imagerie Neurofonctionnelle, Institut des Maladies Neurodégénératives-UMR 5293, CNRS, CEA, University of Bordeaux, Bordeaux, France. [2] Brain Connectivity and Behaviour Laboratory, Sorbonne Universities, Paris, France. [3] Centre for Neuroimaging Sciences, Department of Neuroimaging, Institute of Psychiatry, Psychology and Neuroscience, King's College London, London, UK. [4] Institute of Neurology, UCL, London, UK. ✉email: victor.nozais@gmail.com; michel.thiebaut@gmail.com

Magnetic resonance imaging (MRI) has allowed peering inside the living human brain like never before[1,2]. In particular, functional MRI (fMRI) has allowed the investigation of the brain's dynamic activity and revealed its functional architecture by contrasting the involvement of regions during tasks[3].

While the classical fMRI approach was very fruitful[4], the field of functional neuroimaging has recently moved away from a pure localisationist view of activated brain regions towards an interactive network approach identified with functional[5,6], and effective connectivity[7]. Functional connectivity consists of the coordinated activity of distant brain regions. The resulting brain maps reveal functional networks across brain regions that work in synchrony. However, functional connectivity does not offer any information about the causal relationship between these regions. On the other hand, effective connectivity uses a directional interaction model between brain areas unmasking the modulatory effect some areas have on others within a functional cascade. These methods have demonstrated that the brain is functioning as an interconnected unity rather than a fractionated entity. Despite these advancements, both methods cannot identify the anatomical circuits supporting the interaction between brain regions. Knowing the underlying structural substrates would be crucial, for instance, when the interaction between regions is significant but is supported by an indirect anatomical network. This additional knowledge would have far-reaching implications on the functioning of the healthy brain and in the presence of brain damage.

The brain's anatomical circuits support its functioning[8,9] and are responsible for behavioural and cognitive disorders when impaired[10–13]. Historically, anatomical circuits have been explored in the human brain through post-mortem white matter dissection[14]. Although dissections have enhanced our understanding of the physical connections between brain regions, their use remains limited to post mortem specimens and requires laborious and inherently destructive procedures.

Advances in MRI facilitated the study of the human brain's in vivo circuits through diffusion-weighted imaging tractography[15]. This method measures water diffusion[16], which mainly follows axons' direction, to reconstruct bundles of axons (i.e. pathways). When applied to the entire brain, whole-brain tractography can be computed and is commonly referred to as the structural connectome[17]. This non-invasive in vivo method has been of tremendous help in mapping the anatomy of white matter in healthy[18,19] and clinical populations[10,11,20]. While tractography allows us to study the connections between brain regions, it does not inform us about their functions.

The functional roles of white matter connections have mostly been inferred indirectly by mapping functional deficits on the white matter. This is typically achieved by mapping lesions onto the white matter and scrutinising the resulting functional deficits[21]. However, this approach is limited by the variability of lesion topology and the incomplete mapping of the brain areas contributing to function. Recently, the latest MRI machines' high field strength revealed some moderate task-related white matter fMRI signal[22,23]. While very promising, these results are limited in effect sizes and will need further improvement to generalise this new approach. Additionally, efforts toward statistically linking fMRI structural connectivity and brain function have been undertaken in the past, with both resting-state[24,25] and task-based[26] fMRI. However, these approaches mainly focused on the structural connectivity between grey matter regions rather than the white matter pathways supporting the connectivity. Integrating functional and structural data to explore the function of white matter pathways thus remains a rare endeavour. Only a few studies began to investigate this research line and yielded preliminary evidence demonstrating that it is possible to project fMRI signal onto white matter. Albeit very promising, these attempts have been either limited to resting-state functional connectivity analyses or directly associated task-related functional patterns with the diffusion signal without reconstructing brain circuits[27,28]. Therefore, a method that can directly project task-related fMRI on the white matter is still needed to shed light on the functional role of specific brain circuits.

Here we introduce a method and software: the Functionnectome, that unlocks the function of the white matter. Functional white matter analysis is derived from the combination of task-related functional MRI and new anatomical priors for brain circuits. In doing so, we provide the first functional white matter maps revealing the joint contribution of connected areas to motor, working memory, and language functions. The Functionnectome is available as a companion open-source software (http://www.bcblab.com and https://github.com/NotaCS/Functionnectome).

## Results

In fMRI studies, task-based neural activation patterns are usually derived from the statistical analysis of each voxel's time-course, typically using a general linear model (GLM) with the task's predicted hemodynamic response. To evaluate the participation of white matter pathways in these tasks, we first produced normative anatomical connectivity maps later referred to as priors.

**Priors: anatomical connectivity probability maps**. To project functional signals onto the white matter, the Functionnectome requires prior knowledge of the white matter anatomy, and more specifically, the structural connectivity between a given voxel and the rest of the brain (i.e. priors). Accordingly, each prior is a voxel and its probability of structural connectivity with all other brain voxels. These 3D maps were derived from deterministic tractography of 100 Human Connectome Project (HCP) participants using 7T diffusion-weighted MRI scans[29] that were already processed for tractography by our team[21]. The tractography used is openly available at https://osf.io/5zqwg/. In total, 228,453 maps—one per brain voxel—were generated and are part of the Functionnectome software's priors. The Functionnectome uses the probability indicated in these priors to project the signal from a given voxel to the brain.

A second "simplified" set of priors consisted of 438 cortical[30] and subcortical regions[31] and their probability of structural connectivity with all other brain voxels. We used the second set of "simplified" priors for validation (i.e. comparison with resting-state networks) and possible replication of the analyses in more modest configurations such as laptops.

**Validation of the priors**. To verify the anatomical priors' validity, we used the method developed by O'Muircheartaigh and Jbabdi[32]. Briefly, it compares statistically independent components[33] of resting-state fMRI and structural connectivity. In our analysis, the independent component analysis applied to resting-state fMRI data of the HCP test–retest dataset produced 17 resting-state networks (out of 20 components). Similarly, the independent component analysis applied to the 438 "region-wise" priors produced 50 components.

Cross-correlations indicated that resting-state networks were significantly associated with pairs of structural connectivity components (Fig. 1a). For instance, structural connectivity components 2 and 4 corresponded to the left and right visual resting-state network (RSN2, Fig. 1b) and connected the two hemispheres through the posterior part of the corpus callosum. Likewise, structural connectivity components 37 and 39

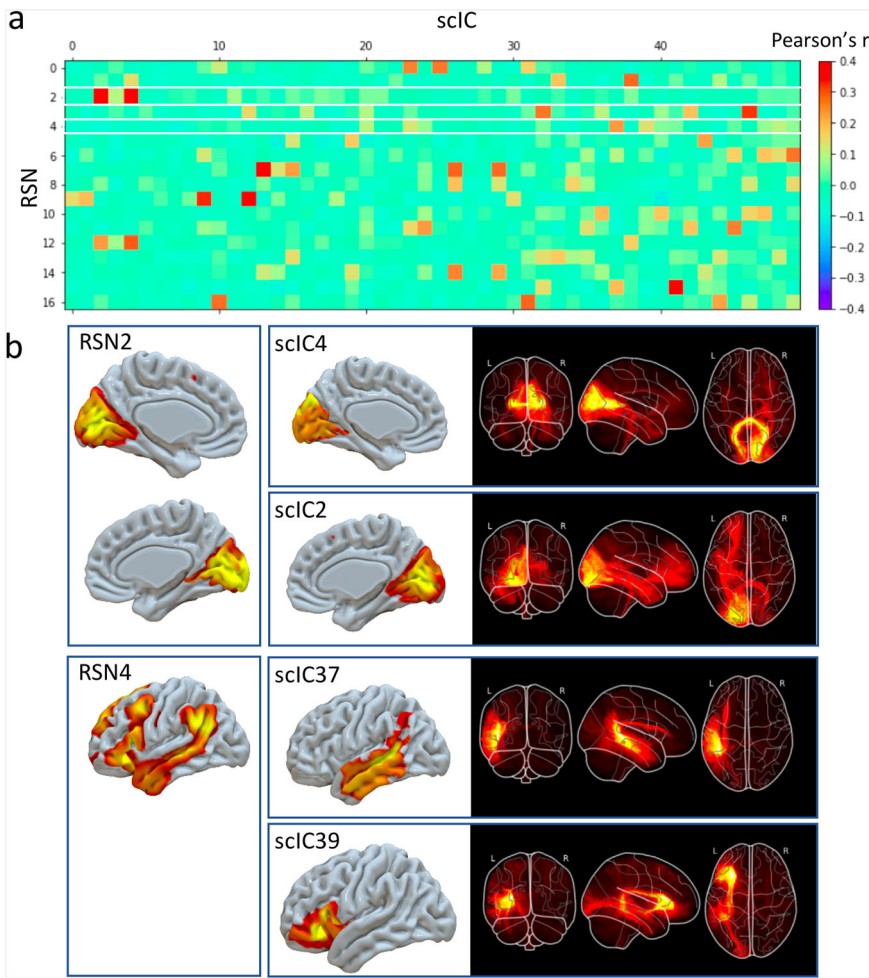

**Fig. 1 Comparison of resting-state networks with structural connectivity independent components.** Top panel (**a**) correlation matrix between the spatial maps of the 17 RSN with the grey matter maps of the 50 scICs. RSN2 and RSN4 are highlighted in white and further described in the bottom panel (**b**) alongside their corresponding scICs. RSN: resting-state networks, scIC: structural connectivity independent components.

corresponded to the language resting-state network (RSN4, Fig. 1b) and included the arcuate fasciculus (the primary language pathway). The good correspondence between classical functional networks and our structural connectivity priors supports their anatomical validity in assessing the probability of connection between functionally activated regions.

**Functionnectome.** Based on these priors, we developed a method, the Functionnectome, which combines the functional signal from distant voxels of the classical fMRI 4D volume (Fig. 2a) using their probabilistic structural relationship given by anatomical priors of the involved brain circuits (Fig. 2b). These priors are probability maps derived from high-resolution tractography (Fig. 2d), indicating the probability of structural connectivity between each grey matter voxel and the rest of the brain. By projecting the fMRI signal from grey matter voxels to the white matter and weighing the signal by the probability of connection, a "functionnectome" (Fig. 2c) is generated. This new 4D volume is compatible with the usual statistical tools to reveal the activation patterns emerging from the signal (Fig. 2e). Here, we illustrated this process by applying the Functionnectome to motor ($n = 46$), working memory ($n = 45$), and language ($n = 44$) functional volumes of the test–retest Human Connectome Project (HCP) dataset. Results from the functionnectome maps are presented side by side with classical task activation analysis. The white matter "activations" displayed on the functionnectome

z-maps correspond to the associated pathways' significant involvement during a task. Importantly, these white matter activations are arising from the projected grey matter signal, and are not direct white matter BOLD signal analysis[22,23]. The white matter activations displayed on the functionnectome z-maps correspond to the associated pathway's significant involvement during a task. Apparent overlaps between white matter activations on the functionnectome maps and grey matter activations on the standard fMRI maps are mostly due to the smoothing applied on standard fMRI.

**Motor tasks.** The right finger taping Functionnectome analysis (Fig. 3a) revealed the well-established motor system circuit with significant involvement of the posterior arm of the left internal capsule, connections to subcortical areas, and the cerebellum through the brain stem (i.e. pons). This activation was accompanied by an involvement of the frontal aslant tract (FAT) connecting the Supplementary motor area (SMA) with the frontal operculum and short U-shaped fibres around the hand area in the primary motor cortex (M1)[34]. The joint contribution of both hemispheres to motor execution[35] was represented by the involvement of the body of the corpus callosum. In contrast, a classical fMRI analysis applied to the same data showed the involvement of the left motor hand area together with the SMA and striatum as well as the right anterior lobe of the cerebellum (Fig. 3b).

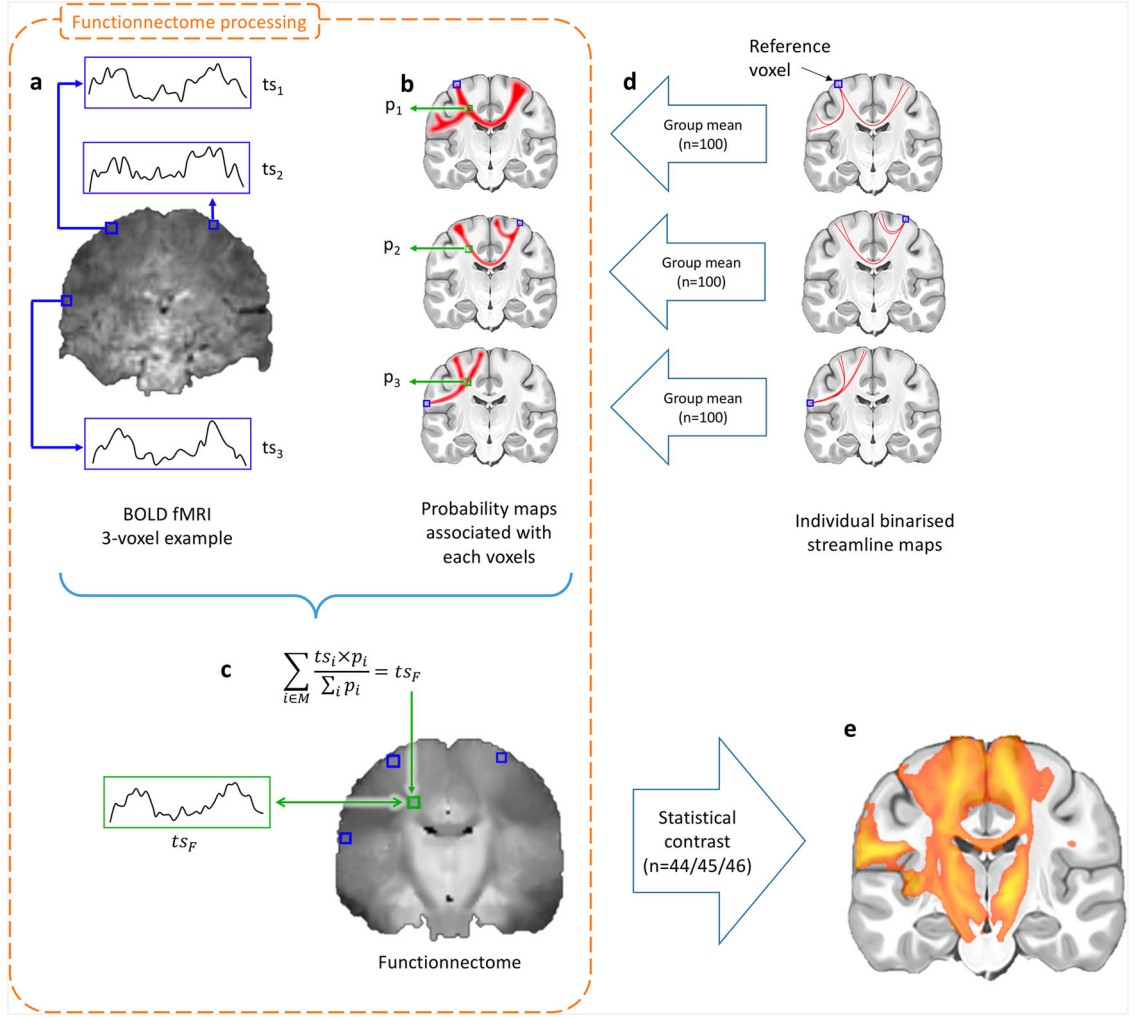

**Fig. 2 Experimental setup of the functionnectome. a** Classical blood-oxygen-level-dependent functional magnetic resonance imaging (BOLD fMRI) in the cortex is projected onto the white matter circuits using (**b**) anatomical priors of the brain circuits and a (**c**) weighted average equation. The priors are derived from (**d**) 100 high-resolution diffusion-weighted imaging datasets. The output of the Functionnectome is subsequently entered (**e**) in the same statistical design as classical fMRI. M: grey matter mask defining which voxels from the input fMRI volume to use in the analysis (which the 3 voxels in **a** would be part of here); ts: time-series, p: the probability of connection.

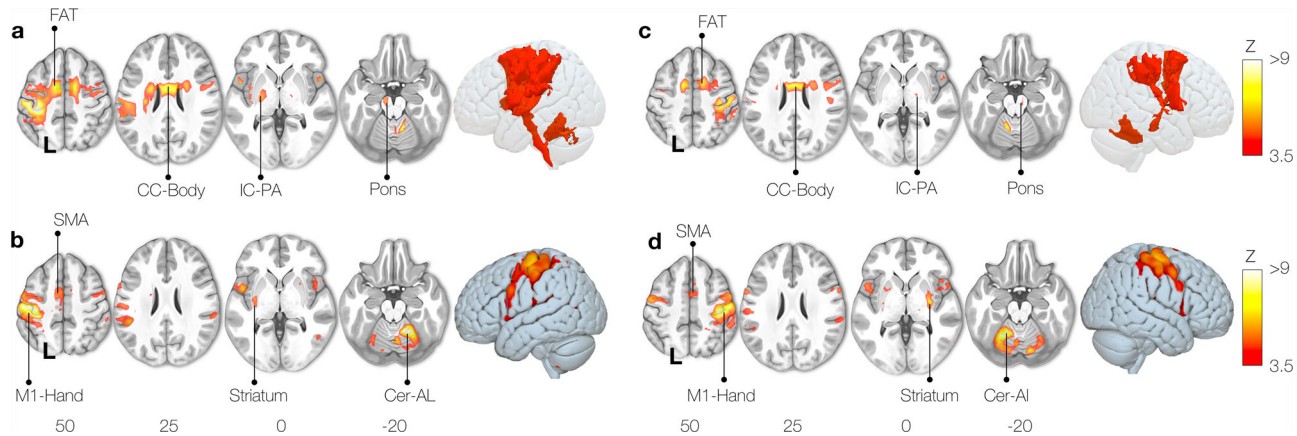

**Fig. 3 Motor task activation network for finger-tapping.** Right hand (**a**: functionnectome, **b**: classic) and left hand (**c**: functionnectome, **d**: classic) finger-tapping activation maps. FAT: Frontal Aslant Tract, CC-Body: Corpus Callosum body, M1: Primary motor cortex, SMA: Supplementary motor area, Cer-AL: Cerebellar anterior lobe, IC-PA: Internal capsule posterior arm.

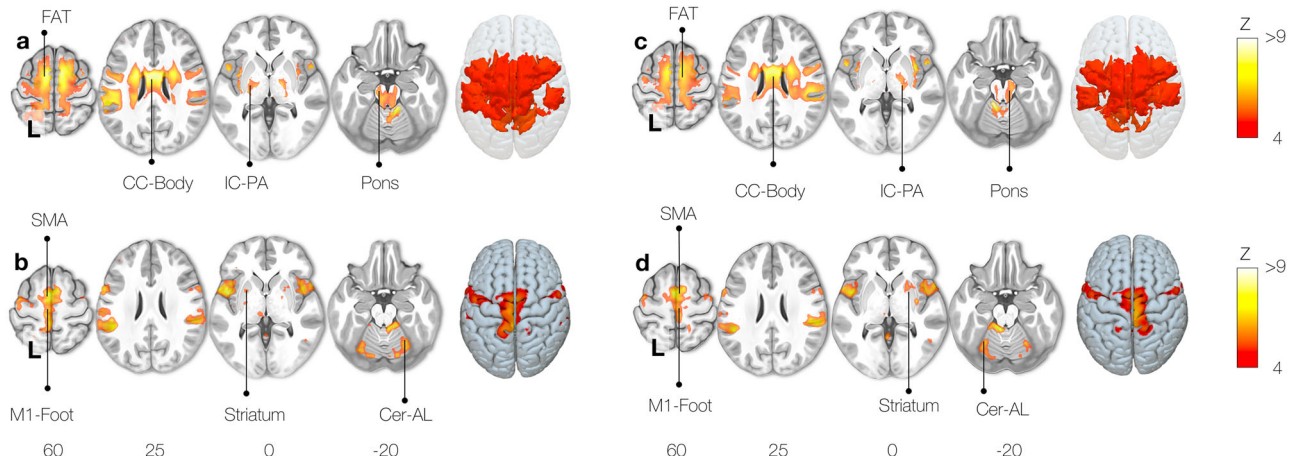

**Fig. 4 Motor task activation network for toe clenching.** Right foot (**a**: functionnectome, **b**: classic) and left foot (**c**: functionnectome, **d**: classic) toe clenching activation maps. FAT: Frontal Aslant Tract, CC-Body: Corpus Callosum body, M1: Primary motor cortex, SMA: Supplementary motor area, Cer-AL: Cerebellar anterior lobe, IC-PA: Internal capsule posterior arm.

These results were elegantly mirrored by the Functionnectome and activation analyses for the left finger tapping task (Fig. 3c–d).

Similarly, the right foot motor activation task (clenching toes) Functionnectome analysis revealed the involvement of the frontal aslant tract, the corpus callosum, the internal capsule, and connections through the pons to the cerebellum (Fig. 4a). Classical analyses showed significant activation of the left motor foot area with the SMA, striatum, and right anterior lobe of the cerebellum (Fig. 4b).

A left-right flipped pattern of results was observed for the left foot motor task Functionnectome and classical analysis (Fig. 4c, d).

**Visual working memory task.** The Functionnectome analysis revealed activation of cortico-cortical circuits necessary for the functional interaction of the frontoparietal areas classically involved in the visual working memory task[36]. As such, the first and second branches of the superior longitudinal fasciculus (SLF 1, 2) were involved (Fig. 5a). SLF1 and SLF2 are linking the superior and middle frontal gyri with the parietal lobe, which are crucial hubs of the working memory network[36]. Likewise, U-shaped fibres, the frontal aslant tract, and the frontal superior longitudinal tract[37] complete the network by connecting different regions within the frontal cortex and the SMA. We also observed the corpus callosum's involvement, which facilitates interhemispheric integration required for working memory[38].

In addition to cortico-cortical pathways, the cortico-ponto-cerebellar tract was also activated and connects the classical fMRI (Fig. 5b), in which disruption has been reported to impair working memory[39].

**Language (semantics).** The Functionnectome analysis for a language task (semantic) is displayed in Fig. 6a and revealed the language "ventral stream" involvement. The ventral stream is composed of the uncinate and the inferior fronto-occipital fasciculi connecting the inferior frontal gyrus with the temporal lobe and the occipital lobe[40–42]. The analysis also showed the middle longitudinal fasciculus, which links the temporal pole to the inferior parietal lobe[43] and the posterior segment of the arcuate fasciculus, which links posterior temporal regions to the parietal lobe[18]. Both hemispheres could interact during the task thanks to the involvement of callosal connections. We also observed significant participation of the fornix during the task.

As expected, the classic fMRI maps displayed activation in the posterior inferior frontal gyrus, the superior temporal gyrus, and the inferior parietal lobe, forming the semantic network[18,44]. The anterior part of the temporal lobe was activated (Fig. 6b), which is in line with the literature highlighting it as a central hub for semantic processing[45]. The orbitofrontal cortex has previously been linked to language and semantic cognition[45]. Lastly, activation of the amygdala, mammillary bodies, and hippocampus reflect a limbic network's involvement, integrating emotions with semantic memory[46].

For each analysis reported above, we repeated the analysis in a replication dataset of the same participants (same session, opposite phase of acquisition). Table 1 indicates the reproducibility rate of the results. The Functionnectome results were systematically more reproducible ($r = 0.82 \pm 0.06$) than the classical analysis ($r = 0.72 \pm 0.05$).

## Discussion

We propose a method, the Functionnectome, to investigate brain circuits' functional involvement during task-related cerebral processes based on openly available anatomical connectivity priors. Applying the Functionnectome to a high-quality functional neuroimaging dataset revealed—for the first time in healthy human brains—the white matter circuits supporting motor, working memory, and language activations. Results also indicated a higher reproducibility of the Functionnectome maps compared to classical task-related activation methods. To support this method's broad uptake and facilitate its application to a wide range of datasets, including in the clinic, we provide a GUI and a terminal-based companion software, as well as simplified priors for faster processing for more modest configurations. This toolbox allows the application of the Functionnectome to any previously acquired fMRI dataset. The toolbox is flexible and users can integrate their own priors and the current release opens up novel avenues for research on the integrative function of white matter.

The Functionnectome incorporates structural connectivity information into the functional analysis, allowing for assessing brain regions' interaction rather than their independent contribution during brain processes. Compared to previous work seeding tractography directly from blobs of functional activations[47], the Functionnectome allows for a statistical assessment of the white matter circuits involved. Additionally, standard fMRI analyses employ spatial filtering before the

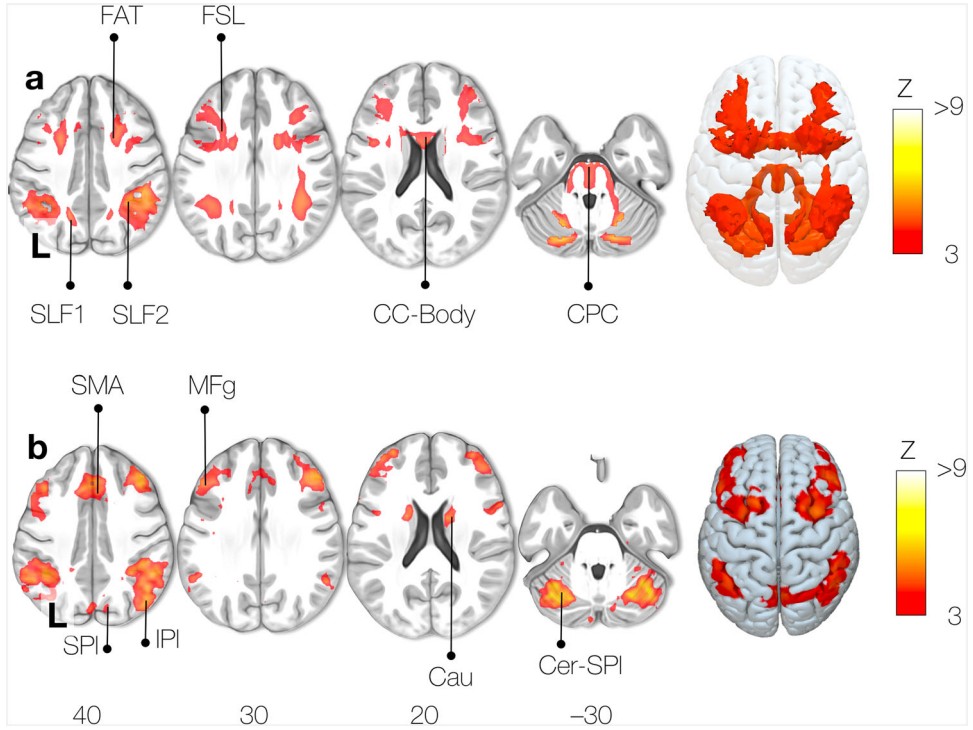

**Fig. 5 Working memory task activation network. a** Functionnectome-derived activation map, (**b**) Classic fMRI activation map. FAT: Frontal Aslant Tract, FSL: Frontal Superior Longitudinal tract, SLF: Superior Longitudinal Fasciculus, CC-Body: Corpus Callosum body. CPC: Cortico-Ponto-Cerebellar tract, SMA: Supplementary Motor Area, SPL: Superior Parietal lobe, IPL: Inferior Parietal lobe, MFg: Middle frontal gyrus, Cau: Caudate nucleus, Cer-SPl: Superior Posterior lobe of the Cerebellum.

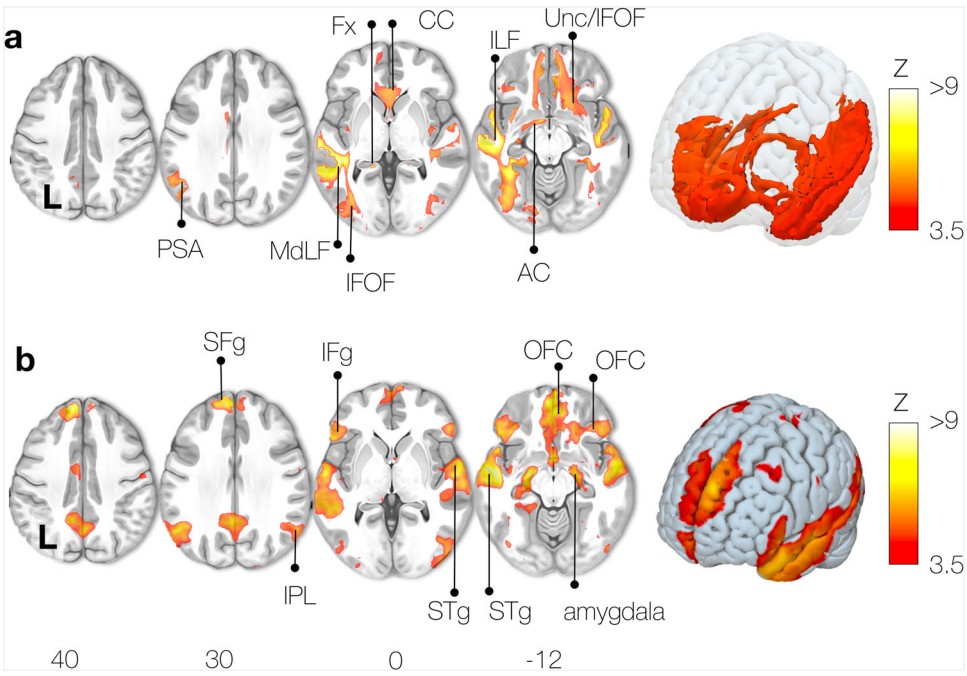

**Fig. 6 Semantic system task activation network. a** Functionnectome-derived activation map, (**b**) Classic fMRI activation map. PSA: Posterior Segment of the arcuate fasciculus, MdLF: Middle Longitudinal Fasciculus, IFOF: Inferior Fronto-Occipital Fasciculus, Fx: Fornix, CC: Corpus Callosum, ILF: Inferior Longitudinal Fasciculus, Unc: Uncinate fasciculus, AC: Anterior Commissure, SFg: Superior Frontal gyrus, IFg: Inferior Frontal gyrus, IPL: Inferior Parietal Lobe, STg: Superior Temporal gyrus, OFC: orbital frontal cortex, hipp hippocampus.

**Table 1 Reproducibility of the results for each task for the Functionnectome and classical fMRI analyses.**

| | Motor | | | | Memory | Language |
|---|---|---|---|---|---|---|
| | Right hand | Left hand | Right foot | Left foot | Working | Semantics |
| **Functionnectome** | 0.78 | 0.73 | 0.82 | 0.85 | 0.84 | 0.89 |
| **Classical fMRI** | 0.71 | 0.68 | 0.70 | 0.72 | 0.70 | 0.83 |

The reproducibility is indicated as Pearson's correlation coefficient *r*.

statistical analysis that is agnostic to the structural connection between the voxels and may mix the signals from functionally unrelated voxels. In contrast, the Functionnectome does not require filtering. As a result, statistical analysis of the functionnectome should reveal, in principle, statistically significant white matter circuits with precision and sensitivity, as discussed in the following sections.

The Functionnectome's priors are based on the current best white matter mapping derived from the Human Connectome Project 7T[29]. The subjacent structure of our priors (i.e. components) demonstrated a good correspondence with resting-state functional networks as previously reported[32]. Despite the limitation of both methods[48,49], the systematic correspondence between the functionnectome's priors and classical functional networks supports their anatomical validity in assessing the probability of connection between functionally activated regions. Furthermore, as resting-state functional connectivity components resemble functional activation networks[50], the similarity between structural connectivity components and resting-state components reaffirms the premise that structural connections of the brain determine its activations and functioning.

With our method, significant activations of brain circuits reflect the functional involvement of brain areas at both ends of the circuit. A white matter pathway links a brain area with another and the signal from both areas is combined. This combination will penalise the statistical activation if only one region is significantly activated by the investigated function. Conversely, if both areas are involved, the combination of their signal in the circuit will promote the statistical detection of this circuit's involvement. As first examples of such functional exploration of white matter, we investigated the brain circuits supporting prominent activation tasks including motor, memory, and language functions. As our knowledge of the fine circuitry of the motor system is mostly derived from animal and lesion studies, the Functionnectome applied to motor tasks offers the unique opportunity to explore these circuits in the healthy human brain. In that regard, our findings match the pathways suggested by the literature and were replicated with both hands and feet twice. We confirmed the involvement of the internal capsule, which is a well-established[51] part of the ascending and descending motor pathways that include the cortico-subcortical motor loops[52,53] and the cortico-cerebellar pathways. The latter has been long known to be part of the motor system[54], with relays in subcortical nuclei, but has never been directly shown in healthy human participants before. The cortico-ponto-cerebellar pathways[55] connect the primary motor cortex of one hemisphere with ipsilateral pontine nuclei and the contralateral anterior lobe of the cerebellum, passing through the internal capsule and the pons. The involvement of the corpus callosum in the motor tasks cannot be directly identified with fMRI but has long been considered essential to integrate the left and right motor systems[56,57]. The Functionnectome maps distinctly isolated these circuits.

Similarly, the patterns of white matter activation from the working memory task confirm and complement the literature. For example, the involvement of the superior longitudinal fasciculus reported with the Functionnectome has been well described[58]. Our results also confirm the importance of the cerebellum in working memory[55,59] but extend this insight by mapping the full circuitry supporting its involvement. In line with this, the fronto-frontal tracts involvement in the working memory functionnectome map supports the hypothesis of the frontal lobe working hierarchically[8,34,60]. Additionally, classical working memory activation tasks[61] and clinical studies[38] unveiled the importance of both hemispheres for working memory, but researchers could only speculate about the underlying anatomical circuitry. Here, the Functionnectome revealed the exact portion of the corpus callosum that integrates bilateral contributions to working memory. Furthermore, the involvement in working memory of the frontal superior longitudinal fasciculus, a fronto-frontal U-shaped pathway, was demonstrated here for the first time. Indeed, this pathway has not been identified in the monkey brain, and its role in working memory in humans has so far only been hypothesised based on its location[37]. This new result highlights the exploratory potential of the Functionnectome toward understanding the white matter support of complex cognitive systems.

The language circuitry, and more so its semantic system, offers an even greater challenge as it cannot be explored in animal studies[41]. The Functionnectome confirmed, for the first time in the healthy human brain, the structural–functional circuitry supporting semantic processes originally suggested by lesion studies[40] or intraoperative stimulation[41] in humans (i.e. the uncinate fasciculus, the inferior fronto-occipital fasciculus, the middle longitudinal fasciculus, and the posterior segment of the arcuate fasciculus). The Functionnectome also revealed the involvement of the anterior corpus callosum in story comprehension. Whilst some language processes require the integration of the left and right hemispheres via the posterior corpus callosum[62], the anterior corpus callosum has been implicated in semantic disorders (e.g. semantic dementia[63]). The Functionnectome result also suggests an involvement of the fornix. As a limbic pathway, the fornix may play an important role in the colouring of the story comprehension with emotions and memories[46]. Our results thus prompt a closer inquiry into the role of the anterior corpus callosum and fornix in semantic processes and offer a non-invasive tool to study its involvement in healthy participants. In sum, our application of the Functionnectome to classical fMRI allows the confirmation and the exploration of the involvement of circuits for specific tasks for the first time in the healthy human brain.

As reproducibility of findings is of utmost importance in science[64], we verified whether our activation maps were consistent across different acquisitions. The replication of our analysis confirmed the high reproducibility of the results highlighted by our method. Importantly, the Functionnectome results were more reproducible than classical task-related activation methods. While the two methods are not identical and not perfectly comparable with regards to filtering, the observed differences also emerge from the fact that they rely on different assumptions for the assessment of the functioning of the brain. While classical

fMRI computes differences between regions of the brain independently, the Functionnectome associates their circuits to brain function. Higher reproducibility for the Functionnectome would then suggest that the functioning of the brain is mediated by interactions via anatomical circuits rather than the isolated contribution of brain regions.

To upscale the validation of this network view of brain functioning, crowdsource analysis of additional data would be possible. To facilitate this application, we provide an open-source software that will allow for an easy use of the Functionnectome method to revisit already acquired fMRI datasets, either privately or publicly available (e.g. HCP[65], UK Biobank[66], ABIDE[67]). The use of the Functionnectome is also not bound to the activation paradigm and can be combined with advanced fMRI statistics[7] to reveal the dynamic causal interaction between brain circuits. Additionally, the Functionnectome can leverage the wealth of MRI modalities to explore the involvement of white matter circuits in different aspects of brain dynamics. For example, it could be applied to resting-state functional connectivity or cortical thickness to open up new perspectives onto the study of functional synchronisation, cortical changes during development, and brain pathologies.

The Functionnectome is a new and promising method that relies on anatomical priors to determine how to project the functional signal onto the white matter. The current best way to obtain a complete 3D map of the white matter pathways in the living human brain is tractography. Although this method has been successfully applied to explore the relationship between white matter structure and brain functions and dysfunctions[11,18], tractography is still facing limitations[48]. For example, a common problem is the difficulty to reconstruct accurately long-range projections[68], which might reduce the probability of detection of the involvement of such pathways with the Functionnectome. Nonetheless, great progress has been made towards the resolution of these problems in the last decade[15]. Future developments in this area will likely improve the quality of current tractograms. These improvements will be implemented in the Functionnectome as priors can easily be replaced in the future to incorporate novel advances in tractography. Future developments of the priors might include a separation into interhemispheric (i.e. commissural circuits), cortico-subcortical (i.e. projection circuits), and cortico-cortical connectivity (i.e. association circuits) to better disentangle brain circuitries.

Finally, the use of the Functionnectome for every brain voxel can be very expensive computationally and thus time-consuming. While we recommend using this procedure, we acknowledge that not all research teams have access to the computing power required to compute the "voxel-wise" Functionnectome over several participants in a reasonable time. To circumvent this potential constraint, we provide an option within the software allowing for the use of an atlas and its parcels instead of all voxels (i.e. 'simplified' priors) for a less computationally intensive "region-wise" analysis (see Supplementary note 1: Region-wise analysis, Supplementary Figs. 1–4 and Supplementary discussion 1 in the supplementary information for the region-wise results and discussion of each of the four tasks explored in the present study).

Overall, we introduced and demonstrated the potential of the Functionnectome method and its open-source companion software (see Supplementary note 2: Functionnectome User-guide in the supplementary information) opening the field of in vivo study of the function of white matter in healthy humans. Despite the unavailability of a ground truth to completely validate the results obtained, we were able to identify strong indicators of both the sensitivity (the expected pathways were detected) and specificity (plausible activation of pathways that are yet to be formally associated with a function, e.g. the FSL fasciculus with working memory) when using the Functionnectome. In this context, the Functionnectome promotes a paradigm shift in the study of the brain, focusing on the interaction of brain regions in the support of a brain function, rather than the fractionated contribution of independent regions.

## Methods
The workflow was summarised in Fig. 7.

**Datasets**. Three datasets derived from the Human Connectome Project[65] (HCP) were used for the study, and are publicly available: the 7T diffusion-weighted imaging (DWI) data used to generate the tractography priors (subset of 100 subjects; available at https://osf.io/5zqwg/ and http://www.bcblab.com/BCB/Opendata.html), the 3 T task-based fMRI acquisitions (46 participants, test–retest dataset), and the 3T resting-state fMRI acquisitions (45 participants, test–retest dataset). In order to ensure that the results obtained here would be generalisable to other datasets, the 100 participants of the DWI dataset were randomly chosen as a normative population that was independent from the participants of the test–retest dataset used in the fMRI analyses. HCP data was acquired by the WU-Minn Consortium with IRB approval and informed consent from all participants, and the WU-Minn HCP Consortium Open Access Data Use Terms were respected in the present study.

**Acquisition parameters and preprocessing**. EPI acquisitions: Full description of the acquisition parameters have been described elsewhere[69]. In brief, the data were acquired on a 3 Tesla Siemens Skyra scanner using a whole-brain EPI acquisition with a 32-channel head coil. The parameters were as follows: TE = 33.1 ms, TR = 720 ms, flip angle = 52°, BW = 2290 Hz/Px, in-plane FOV = 208 × 180 mm, 72 slices, 2.0 mm isotropic voxels, and a multi-band acceleration factor of 8. 1200 frames per acquisition for the resting-state fMRI, and the number of frames was task-dependent for the task-based fMRI. Each type of acquisition was acquired twice using a right-to-left and a left-to-right phase encoding. The EPI 4D acquisitions were then preprocessed through the "Minimal preprocessing pipeline" *fMRIVolume*[70], which applies movement and distortion corrections and performs a registration to the MNI152 space. Additionally, the resting-state acquisitions were further preprocessed with despiking, detrending of motion and CSF, white matter and grey-matter signal, temporal filtering (0.01–0.1 Hz), and spatial smoothing (5 mm FWHM).

**Diffusion-Weighted Imaging (DWI) acquisitions and tractography**. Structural connectome data were downloaded (subset of 100 subjects; available at https://osf.io/5zqwg/ and http://www.bcblab.com/BCB/Opendata.html). This dataset was derived from the diffusion-weighted imaging dataset of 100 participants acquired at 7 Tesla by the Human Connectome Project Team[29] (http://www.humanconnectome.org/study/hcp-young-adult/; WU-Minn Consortium; Principal investigators: David Van Essen and Kamil Ugurbil; 1U54MH091657).

The scanning parameters have previously been described in ref. [29]. In brief, each diffusion-weighted imaging consisted of a total of 132 near-axial slices acquired with an acceleration factor of 3[71], isotropic (1.05 mm³) resolution and coverage of the whole head with a TE of 71.2 ms and with a TR of 7000 ms. At each slice location, diffusion-weighted images were acquired with 65 uniformly distributed gradients in multiple Q-space shells[72] and 6 images with no diffusion gradient applied. This acquisition was repeated four times with a *b*-value of 1000 and 2000 s/mm$^{-2}$ in pairs with anterior-to-posterior and posterior-to-anterior phase-encoding directions. The default HCP preprocessing pipeline (v3.19.0)[70] was applied to the data[73]. In short, the susceptibility-induced off-resonance field was estimated from pairs of images with diffusion gradient applied with distortions going in opposite directions[74] and corrected for the whole diffusion-weighted dataset using TOPUP[75]. Subsequently, motion and geometrical distortion were corrected using the EDDY tool as implemented in FSL.

Next, we discarded the volumes with a *b*-value of 1000 s/mm$^{-2}$ and whole-brain deterministic tractography was subsequently performed in the native DWI space using StarTrack software (https://www.mr-startrack.com). A damped Richardson–Lucy algorithm was applied for spherical deconvolutions[75,76]. A fixed fibre response corresponding to a shape factor of $\alpha = 1.5 \times 10^{-3}$ mm$^2$ s$^{-1}$ was adopted, coupled with the geometric damping parameter of 8. Two hundred algorithm iterations were run. The absolute threshold was defined as three times the spherical fibre orientation distribution (FOD) of a grey matter isotropic voxel and the relative threshold as 8% of the maximum amplitude of the FOD[77]. A modified Euler algorithm[78] was used to perform whole-brain streamline tractography, with an angle threshold of 35°, a step size of 0.5 mm and a minimum streamline length of 15 mm.

We co-registered the structural connectome data to the standard MNI 2 mm space using the following steps: first, whole-brain streamline tractography was converted into streamline density volumes where the intensities corresponded to the number of streamlines crossing each voxel. Second, a study-specific template of streamline density volumes was generated using the Greedy symmetric

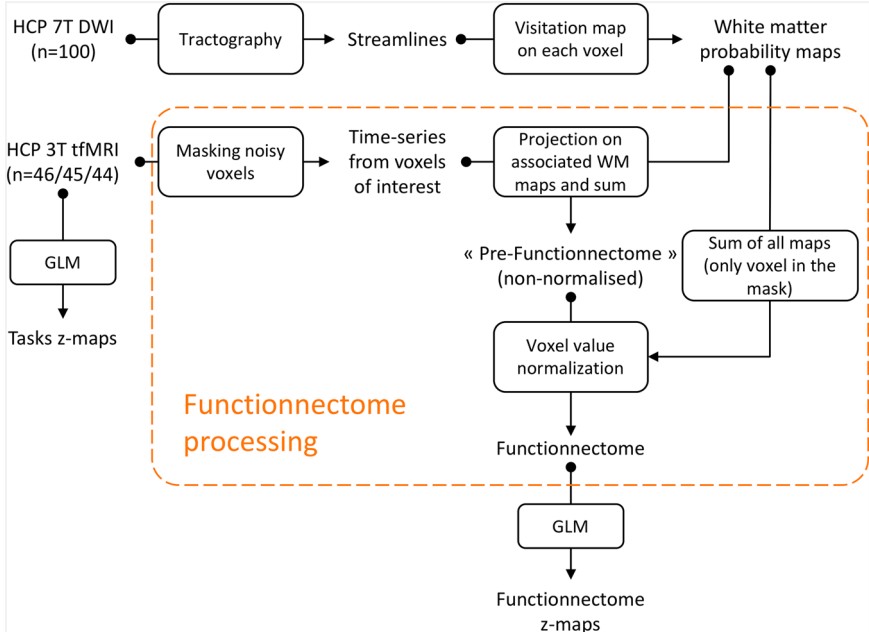

**Fig. 7 Graphical summary of the neuroimaging workflow.** Dashed orange lines delimitate the original part of the processing workflow called "Functionnectome" from other classical processing. Arrows indicate the direction of the workflow. Boxes correspond to the processing steps.

diffeomorphic normalisation (GreedySyN) pipeline distributed with ANTs[78,79]. This provided an average template of the streamline density volumes for all subjects. The template was then co-registered with a standard 2 mm MNI152 template using flirt as implemented in FSL. This step produced a streamline density template in the MNI152 space. Third, individual streamline density volumes were registered to the streamline density template in the MNI152 space template and the same transformation was applied to the individual whole-brain streamline tractography using the trackmath tool distributed with the software package Tract Querier[80] using ANTs GreedySyn. This step produced a whole-brain streamline tractography in the standard MNI152 space.

**Tractogram-derived anatomical priors**. Our method projects the functional signal to white matter structures. To do so, knowledge of the underlying brain circuits is required. For each voxel of the brain, we produced a map indicating the probability of structural connectivity between the said voxel and all the other voxels of the brain. These maps were derived from the tractography results of a normative population of 100 HCP participants. Specifically, for each voxel, all streamlines intersecting this specific voxel were derived from tractograms, binarised and averaged across subjects, using the "*Disconnectome*" function of the BCBtoolkit[81]. Thus, each map has voxel values between 0 and 1, representing the probability of the presence of a streamline, thus offering a proxy at the population level of the probability of connection between white matter voxels and a reference voxel[82].

Additionally to these voxel-wise priors (each voxel associated with a probability map), region-wise probability maps were generated using the same approach: to build the probability map of a region, all streamlines passing through this region were used (instead of a single voxel). Here we used the HCP multi-modal parcellation (MMP)[30] of the brain as the reference together with subcortical areas derived from AAL3[31].

These probability maps serve as the anatomical priors for brain circuits. As part of the toolbox, the priors can be adjusted in the future.

**Comparing anatomical priors and resting-state networks**. We followed the method developed by Jonathan O'Muircheartaigh and Jbabdi[32] to compare cortical maps derived from functional and structural connectivity. The functional connectivity maps here are the so-called resting-state networks (RSN), composed of brain regions displaying synchronous brain activity during rest. Independent Component Analysis (ICA) is a popular data-driven method capable of extracting these networks from the resting-state fMRI data by detecting spatially independent components in the signal, and isolating their time-courses and corresponding spatial maps. We first ran a group ICA on the resting-state data (both LR and RL acquisition of all 45 subjects) to extract 20 independent components (IC), using MELODIC (multivariate exploratory linear optimised decomposition into independent components, version 3.15), available in the FMRIB Software Library (FSL)[75]. Among these 20 functional ICs, 3 were identified as noise components, leaving 17 ICs as resting-state networks. In parallel, we performed the same ICA on the anatomical priors of the Functionnectome using the region-wise probability maps concatenated over a fourth dimension. The ICA was thus driven by the

patterns of structural connectivity emerging from the priors, effectively grouping the white matter pathways into homogeneous ICs. We extracted the grey matter pattern for each RSN and for each structural connectivity ICs using standard subcortical areas[31] and cortical areas derived from a multimodal atlas of the brain surface[30], zeroing the negative part of the maps. Similarity between both sets of maps was subsequently assessed by means of Pearson correlation.

**Functionnectome pre-processing**. The Functionnectome method projects the BOLD signal obtained for each grey matter voxel onto the related white matter voxels. The whole principle of the method is akin to a weighted average, on a given voxel, of the BOLD signal from the voxels sharing a structural link (given by the anatomical priors) to this voxel. Thus, the mathematical formulation of the concept can be summed up by the following equation:

For a given voxel $v$ (spatial coordinates) at a given time-point $t$, the value of that voxel in a functionnectome is

$$Functionnectome(v, t) = \sum_{m \in M} \frac{P_m(v) \times F(m, t)}{\sum_{m \in M} P_m(m, t)}$$

With $M$ the set of voxels selected by the input mask; $P_m$ the probability map derived from the voxel $m$; and $F$ the original fMRI 4D volume.

In other words, if we focus on a single functionnectome voxel $v$, the value of this voxel is equal to the sum of the BOLD signal from every voxel in the brain, weighted by the probability of their connection to $v$ (which is 0 if they are not part of the involved circuit), and divided by the sum of all those probabilities. This last step of division ensures that all voxels have the same range of values as the classical grey-matter BOLD signal. Without this step, the signal of the resulting functionnectome would not be homogeneous over the brain, with, for example, voxels in dense white matter circuits showing signals of higher amplitude than the rest of the brain.

Practically, the algorithm used to apply this method follow these few steps:

For each subject, the Functionnectome was provided with a mask selecting the voxels whose functional signal will be projected onto the white matter. Here we used the masks available from the HCP that excluded noisy voxels. These masks exclude voxels with a high coefficient of variation, i.e. higher than 0.5 standard deviations compared to neighbouring voxels (saved by the HCP pipeline in the file "RibbonVolumeToSurfaceMapping/goodvoxels.nii.gz").

Next, the time-series was extracted for each voxel and multiplied with its associated probability map. In doing so the functional signal is projected on the white matter, and weighted by the probability of the presence of a streamline, resulting in one 4D volume per voxel. All 4D volumes are subsequently fused together by voxel-wise addition and divided by the sum of all probability maps to produce a weighted average of the voxel-wise 4D volumes. This final step ensures that all voxels have a comparable range of values (equivalent to the range of values of the BOLD signal). The final output of the algorithm is a functionnectome 4D volume.

**Functional data pre-processing**. To compare our method with standard analyses, we explored the activation patterns of HCP fMRI paradigms for motor ($n = 46$), working memory ($n = 45$), and language ($n = 44$) tasks (see https://protocols.humanconnectome.org/HCP/3T/task-fMRI-protocol-details.html and ref. [83] for a full description of the tasks protocol). Briefly, the motor tasks consisted of finger tapping (left or right hand) and toes clenching (left or right foot); the working memory task was a 2-back task; and the language task corresponded to the comparison of comprehension of high and low semantic content (stories, math problems).

**Processing of neuroimaging data**. The activation analysis was applied to both the original functional dataset and the functionnectome 4D volumes. Processing was identical except for the application of a traditional spatial smoothing (FWHM = 4 mm) on the functional dataset (i.e. as a usual step to improve the signal/noise ratio and misalignment). Specifically, no spatial smoothing is required for the functionnectomes 4D volumes. Usual smoothing aims at improving the signal/noise ratio (SNR) using a weighted average of the local signal, assuming that neighbouring voxels share some signal of interest. The functionnectome method combines the signal from distant yet structurally linked voxels, which has an analogous effect of improving the SNR, but is guided by actual brain circuits.

The processing was done using FEAT (FMRI Expert Analysis Tool, v6.00) from FSL (FMRIB's Software Library). First-level statistical analysis was carried out on each participant using FILM (FMRIB's Improved Linear Model)[84] with prewhitening. For each task, we used the main contrasts provided with the HCP dataset. Then, a group-level analysis was performed on all participants using the first stage of FLAME (FMRIB's Local Analysis of Mixed Effects)[85], resulting in the z-maps (z-transformed t-maps) presented in the results (Figs. 1–4). Note that the assumption of identically independently distributed residuals in the linear modelling applied to the functionnectome volumes should hold true, as the signal of the voxels results from a simple linear combination of the signals (the classic BOLD time-series) for which the assumption was already considered valid.

**Statistics and reproducibility**. To test the reproducibility of our results, we used the two acquisitions realised for each subject: one with a left-right encoding phase (main analysis), the other with a right-left encoding phase (reproducibility analysis). As the reproducibility analysis used the data from the same original subjects, the sample size was the same as the one of the main analysis, with $n = 46$ for the motor task, $n = 45$ for the working memory task, and $n = 44$ for the language task. We compared the z-maps resulting from the full processing of these two acquisitions using Pearson's correlation coefficient (excluding voxels outside the brain).

**Visualisation**. Functional z maps and functionnectome maps were displayed on a standard template in MRIcron (https://www.nitrc.org/projects/mricron). Labelling for cortical regions and white matter pathways were added manually by expert anatomists (SJF and MTS). The visualisation of 3D structures in 2D is limited and may appear ambiguous at times but the full trajectories of pathways were considered before labelling, especially in regions of overlap between white matter structures. The 3D renderings were generated using the associated z-map. All functionnectome maps (slices and 3D renderings) were masked to remove the grey-matter parts of the volume in order to improve readability of the figures. The mask used here was generated using the segmentation provided in the HCP dataset and was composed of the voxels defined as white matter or brain stem in at least 10% of the subjects of the test–retest dataset. This very permissive 10% threshold was chosen to prevent the underestimation of the extent of the functionnectome maps. Note that the grey-matter parts of the maps are also of interest, and were only removed here to avoid confusion between the two methods.

**Availability of data**. All the raw anatomical and functional data are available on the HCP website. The Functionnectomes and the associated maps are available on demand to the authors. The python compatible algorithm (version 3.6 or higher) for the Functionnectome analysis is freely available and comes with an optional GUI code.

**Reporting summary**. Further information on research design is available in the Nature Research Reporting Summary linked to this article.

## Data availability
All the raw anatomical and functional data are available on the HCP website.
https://www.humanconnectome.org

## Code availability
The Functionnectome and the associated maps are available on demand to the authors. The python compatible algorithm (version 3.6 or higher) for the Functionnectome analysis is freely available and comes with an optional GUI code (https://github.com/NotACS/Functionnectome and http://www.bcblab.com).

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

## Acknowledgements

We thank the University of Bordeaux and CNRS for the infrastructural support. This project has received funding from the European Research Council (ERC) under the

European Union's Horizon 2020 research and innovation programme (Grant agreement No. 818521) and the Marie Skłodowska-Curie programme (Grant agreement No. 101028551). Data were provided by the Human Connectome Project, WU-Minn Consortium (Principal Investigators: David Van Essen and Kamil Ugurbil; 1U54MH091657) funded by the 16 NIH Institutes and Centers that support the NIH Blueprint for Neuroscience Research; and by the McDonnell Center for Systems Neuroscience at Washington University.

## Author contributions

V.N. implemented the methods, performed the analyses, and wrote the manuscript. S.J.F wrote the manuscript and reviewed the neuroimaging data. C.F. conceived the study, implemented part of the methods and revised the manuscript. L.P. reviewed the neuroimaging data and revised the manuscript. M.T.S. conceived and coordinated the study, implemented the methods, performed the analyses, wrote the manuscript, and provided funding.

## Competing interests

The authors declare no competing interests.
