## [Peer Review File · Communications Biology]

Reviewers' comments:

Reviewer #1 (Remarks to the Author):

Sorry for the delay.

This is a very insightful paper written from a prominent group in neuroimaging. They have described a novel way to bridge together the complimentary approaches into a single framework. The paper has been very well written, methods adequately described, and results and discussion very well presented.

I had a few minor points.

1. Was the same subjects used for both functional connectivity and DTI results. While the authors do describe 100 subjects, it was not clear to me if the same subjects were used.
2. Why did the authors choose the DTI data from a 7T and resting state data from 3T. It would help to clarify.
3. The resting state state analysis, even though briefly should be described in the document.

Reviewer #2 (Remarks to the Author):

The paper presents a computational method to combine brain fMRI activation maps with structural connectivity maps (from dMRI). The goal is to create whole brain maps that jointly take into account the main white matter (WM) pathways, supporting a given brain function, and gray matter (GM) areas BOLD signal. The authors refer to this process as "projecting" the cortical/subcortical activations onto the WM, in order to create maps akin to WM BOLD activation but in fact different (as noted in the last sentence of page 8) as they are closer to a virtual entity (called "Functionnectome") that can provide a new "contrast" for brain areas both functionally and structurally "activated". The new method is extensively illustrated by applying it to the HCP motor, working memory and language task fMRI data.

This is a very interesting new method, with significant potential for the neuroimaging community. The paper is clear and well-written. The method appears to be technically sound and the toolbox is importantly available as open source software. The experiments are convincing and support the conclusions. There are however several important points that I believe need to be addressed.

1. It is reasonable to assume that the techniques used to generate the anatomical connectivity prior (at the core of the proposed method) have a significant impact on the overall results. Therefore, I believe it is important to assess the influence of the fiber orientation mapping and tractography methods. Specifically, the authors should demonstrate the stability (similar to Table 1) of their results when using probabilistic tractography (e.g. probtractx) and other ODF estimation methods which can handle the full HCP dMRI data (i.e. avoid discarding the $b=1000\text{s/mm}^2$ shell).

Related to this point, it is well known that tractography uncertainty increases as a function of the length of the underlying WM pathway. How sensitive is the method to this? I believe there is a bias that will affect WM "activations" more substantially for long-range connections. This should at least be discussed.

2. It is not clear what M (Fig 2) is. It is indeed described in the Methods section but it should be illustrated on Fig 2.

3. It was chosen to downsample the structural connectivity data (originally at 1.05mm^3) to 2.0mm^3 , presumably to match the resolution of the fMRI data. This seems like it would lead to a significant loss of information. Why couldn't the method handle different resolutions for fMRI and dMRI data (which is very common) and reconstruct the Functionnectome at the dMRI resolution (usually higher)?

4. P 16. It is noted that the method "requires the functional involvement of the brain areas at both ends of the circuit". Could the authors elaborate on that, as it is not clear why this is technically required?

Related to this, since structural connectivity is computed for all brain locations, was symmetry enforced? Would it be advantageous to do so?

5. Finally, Figs. 3,4,5,6 show that, compared to the standard analysis (BOLD), the proposed method can decrease/increase the spatial extent of "activations" in the (sub)cortical GM (e.g. Cer-AL on Fig 3 and IC-PA on Fig 4). How can this be interpreted since it is in the GM, where the BOLD signal is presumably more accurate? Is this a limitation of the method?

Reviewer #1

This is a very insightful paper written from a prominent group in neuroimaging. They have described a novel way to bridge together the complimentary approaches into a single framework. The paper has been very well written, methods adequately described, and results and discussion very well presented.

We thank you for your very positive assessment.

1. Was the same subjects used for both functional connectivity and DTI results. While the authors do describe 100 subjects, it was not clear to me if the same subjects were used.

We thank the reviewer for pointing out the need for clarification. The 100 participants used to generate the probability maps are indeed not the same as the ones used in the functional analyses (both task fMRI and resting-state fMRI). The 100 participants were chosen as a normative population, and to validate the generalisability of the probability maps (which are provided with the software as anatomical priors). Therefore, using a different dataset for the functional analyses was deemed necessary for other teams to be able to use our software and analyses with their own fMRI data (even in the absence of available diffusion-weighted imaging). We updated the methods to clarify this point:

“In order to ensure that the results obtained here would be generalisable to other datasets, the 100 participants of the DWI dataset were randomly chosen as a normative population that was independent from the participants of the test-retest dataset used in the fMRI analyses.”

2. Why did the authors choose the DTI data from a 7T and resting state data from 3T. It would help to clarify.

We thank the reviewer from bringing up this important point. We used the 7T DWI dataset as it provides the best spatial resolution (1.05 mm³ voxels), and thus the best tractography results for the creation of the probability maps. As for the resting-state analysis, we used the same participants as in the task fMRI analysis for consistency. The test-retest dataset of the HCP does not have 7T resting-state scans.

3. The resting state analysis, even though briefly should be described in the document.

We have expanded the method section to include more details about the resting-state analysis, and clarify the principle of the Independent Component Analysis used here:

“...maps derived from functional and structural connectivity. The functional connectivity maps here are the so-called resting-state networks (RSN), composed of brain regions displaying synchronous brain activity during rest. Independent Component Analysis (ICA) is a popular data-driven method capable of extracting these networks from the resting-state fMRI data by detecting spatially independent components in the signal, and isolating their time-courses and corresponding spatial maps. We first ran a group ICA...”

Reviewer #2

The paper presents a computational method to combine brain fMRI activation maps with structural connectivity maps (from dMRI). The goal is to create whole brain maps that jointly take into account the main white matter (WM) pathways, supporting a given brain function, and gray matter (GM) areas BOLD signal. The authors refer to this process as “projecting” the cortical/subcortical activations onto the WM, in order to create maps akin to WM BOLD activation but in fact different (as noted in the last

sentence of page 8) as they are closer to a virtual entity (called “Functionnectome”) that can provide a new “contrast” for brain areas both functionally and structurally “activated”. The new method is extensively illustrated by applying it to the HCP motor, working memory and language task fMRI data.

This is a very interesting new method, with significant potential for the neuroimaging community. The paper is clear and well-written. The method appears to be technically sound and the toolbox is importantly available as open source software. The experiments are convincing and support the conclusions. There are however several important points that I believe need to be addressed.

Thank you for your very positive feedback.

1. It is reasonable to assume that the techniques used to generate the anatomical connectivity prior (at the core of the proposed method) have a significant impact on the overall results. Therefore, I believe it is important to assess the influence of the fiber orientation mapping and tractography methods. Specifically, the authors should demonstrate the stability (similar to Table 1) of their results when using probabilistic tractography (e.g. probtractx) and other ODF estimation methods which can handle the full HCP dMRI data (i.e. avoid discarding the $b=1000\text{s/mm}^2$ shell). Related to this point, it is well known that tractography uncertainty increases as a function of the length of the underlying WM pathway. How sensitive is the method to this? I believe there is a bias that will affect WM “activations” more substantially for long-range connections. This should at least be discussed.

We chose to use deterministic tractography to build the probability maps as anatomical studies have shown its reliability (Catani et al., 2012; Dell’Acqua et al., 2013; Karolis et al., 2019; Thiebaut de Schotten et al., 2011a). Specifically, the parameters used in the tractography algorithm were chosen for their reliability regarding results with axonal tracing studies and with post-mortem dissections. Moreover, differences that might arise from the comparison between analyses with different probability maps would only reflect the differences between the tractography methods. While such comparison is interesting it is beyond the scope of the present study that provide the software to carry out the analysis and the mean for users to build their own priors. Optimisation of the priors is a complex research that goes well beyond probabilistic/deterministic algorithms and is computationally exceptionally intensive. A “challenge” type approach, as presented by Maier-Hein and colleagues in 2017, will be required in order to comprehensively compare parameters. We thank the reviewer for this suggestion that will be the subject of future work from our team and other teams who would like to use our open software. We now edited the user guide to encourage the scientific community to build and test their own priors.

The reconstruction of long-range WM pathways is a bias common to all types of tractography methods (Thomas et al., 2014) that we now acknowledge in the discussion as follows:

“... tractography is still facing limitations. For example, a common problem is the difficulty to reconstruct accurately long-range projections (Thomas et al., 2014), which might reduce the probability of detection of the involvement of such pathways with the Functionnectome.”

Catani et al., 2012 : <https://doi.org/10.1016/j.cortex.2011.12.001>

Dell’Acqua et al., 2013 : <https://doi.org/10.1002/hbm.22080>

Karolis et al., 2019 : <https://doi.org/10.1038/s41467-019-09344-1>
Thiebaut de Schotten et al., 2011a : <https://doi.org/10.1038/nn.2905>
Thomas et al., 2014 : <https://doi.org/10.1073/pnas.1405672111>
Maier-Hein et al., 2017 : <https://doi.org/10.1038/s41467-017-01285-x>

2. It is not clear what M (Fig 2) is. It is indeed described in the Methods section but it should be illustrated on Fig 2.

We thank the reviewer for identifying the omission. We completed the description of M in the legend of Figure 2:

“M: grey matter mask defining which voxels from the input fMRI volume to use in the analysis”

3. It was chosen to downsample the structural connectivity data (originally at 1.05mm³) to 2.0mm³, presumably to match the resolution of the fMRI data. This seems like it would lead to a significant loss of information. Why couldn't the method handle different resolutions for fMRI and dMRI data (which is very common) and reconstruct the Functionnectome at the dMRI resolution (usually higher)?

We thank the reviewer for suggesting this possible improvement to our method. We want to emphasise that the tractography was computed at the original 1.05mm³ resolution, and was only downsampled to the fMRI resolution in the final step, when creating the probability maps. Concerning the possibility to use the dMRI resolution in the Functionnectome, we acknowledge that this could improve the final results by showing finer white matter structures, but at the cost of significantly increasing the computational load of the method (which is already quite high) as well as the correction for multiple comparisons. Nevertheless, we are now planning to add this option to the software in a later release. Thank you for your suggestion.

4. P 16. It is noted that the method “requires the functional involvement of the brain areas at both ends of the circuit”. Could the authors elaborate on that, as it is not clear why this is technically required?

We thank the reviewer for highlighting this lack of clarity. The term “requires” was too strong and the concept was not properly explained in this section. The functional signal from a brain region (or more generally any voxel) is projected to all white matter voxels structurally linked to this voxel, as suggested by the probability maps. White matter voxels of the fonctionnectome volume then display a mix of the signal from multiple brain regions. During the statistical analysis, this mixing will penalise the detection of the contrast in white matter voxels with little projections from the classically activated areas (e.g. in pathways where only one end of the circuit is activated), and will conversely maximise the detection in circuit with the functional involvement of the brain areas at both ends.

The manuscript was amended with the following paragraph in the discussion:

“With our method, significant activations of brain circuits reflect the functional involvement of brain areas at both ends of the circuit. A white matter pathway links a brain area with another and the signal from both areas is combined. This combination will penalise the statistical activation if only one region is significantly activated by the investigated function. Conversely, if both areas are involved, the combination of their signal in the circuit will promote the statistical detection of this circuit's involvement.”

Related to this, since structural connectivity is computed for all brain locations, was symmetry enforced? Would it be advantageous to do so?

No symmetry criterion was enforced at any point in the processing of the data. The structural connectivity probability maps aim at reflecting the true (albeit average across the 100 participants) white matter connectivity of the brain. As it has been demonstrated that the white matter displays hemispheric asymmetries (Thiebaut de Schotten et al. 2011b, Lawes et al., 2008), enforcing such symmetry could reduce the anatomical significance of the activation, or even lead to spurious activation due to misalignments between grey matter activation and white matter circuits.

Thiebaut de Schotten 2011b : <https://doi.org/10.1016/j.neuroimage.2010.07.055>

Lawes 2008: <https://doi.org/10.1016/j.neuroimage.2007.06.041>

5. Finally, Figs. 3,4,5,6 show that, compared to the standard analysis (BOLD), the proposed method can decrease/increase the spatial extent of “activations” in the (sub)cortical GM (e.g. Cer-AL on Fig 3 and IC-PA on Fig 4). How can this be interpreted since it is in the GM, where the BOLD signal is presumably more accurate? Is this a limitation of the method?

We thank the reviewer for his/her question.

If we understand the reviewer’s comment correctly, the question is why, on some of the displayed slices in the figures, the spatial extent of the activation is bigger/smaller on the grey matter maps than on the grey matter part of the fonctionnectome maps.

The fonctionnectome maps displayed on the top panel in the figures 3, 4, 5, and 6 are showing white matter activations only. It is important to stress that the fonctionnectome activations are in the white matter and sometimes partially overlap with standard fMRI activations mostly due to the smoothing employed with standard fMRI. To clarify this, we edited the “Fonctionnectome” paragraph of the results with the following content:

“The white matter activations displayed on the fonctionnectome z-maps correspond to the associated pathway’s significant involvement during a task. Apparent overlaps between white matter activations on the fonctionnectome maps and grey matter activations on the standard fMRI maps are mostly due to the smoothing applied on with standard fMRI.”

If, however, the comment was about the difference in size of the activations between the standard and the fonctionnectome analysis, it is due to the geometry of the white matter structures involved. Indeed, due to the three-dimensional characteristic of white matter pathways, some axial slices will display smaller or bigger activations when compared to traditional BOLD. Also, white matter pathways involved in a function will necessarily appear larger than their grey matter counterpart as they must connect distant brain areas activated by the investigated function.

REVIEWERS' COMMENTS:

Reviewer #1 (Remarks to the Author):

The authors have done a good job addressing the questions and concerns raised. I have no additional comments.

Reviewer #2 (Remarks to the Author):

The authors have appropriately addressed my questions and comments. Thank you.